# STAR-VP: Improving Long-term Viewport Prediction in 360° Videos via Space-aligned and Time-varying Fusion

### Baoqi Gao
State Key Laboratory of Networking
and Switching Technology, Beijing
University of Posts and
Telecommunications
Beijing, China
E-BYTE Technology Co., Ltd.
Beijing, China
gaobaoqi@bupt.edu.cn

### Daoxu Sheng
State Key Laboratory of Networking
and Switching Technology, Beijing
University of Posts and
Telecommunications
Beijing, China
E-BYTE Technology Co., Ltd.
Beijing, China
shengdx@bupt.edu.cn

### Lei Zhang
China Unicom Network
Communications Co., Ltd.
Beijing, China
zhangl83@chinaunicom.cn

### Qi Qi[*]
State Key Laboratory of Networking
and Switching Technology, Beijing
University of Posts and
Telecommunications
Beijing, China
qiqi8266@bupt.edu.cn

### Bo He
State Key Laboratory of Networking
and Switching Technology, Beijing
University of Posts and
Telecommunications
Beijing, China
hebo@bupt.edu.cn

### Zirui Zhuang
State Key Laboratory of Networking
and Switching Technology, Beijing
University of Posts and
Telecommunications
Beijing, China
zhuangzirui@bupt.edu.cn

### Jingyu Wang[*]
State Key Laboratory of Networking
and Switching Technology, Beijing
University of Posts and
Telecommunications
Beijing, China
wangjingyu@bupt.edu.cn

## ABSTRACT

Accurate long-term viewport prediction in tile-based 360° video adaptive streaming helps pre-download tiles for a further future, thus establishing a longer buffer to cope with network fluctuations. Long-term viewport motion is mainly influenced by Historical viewpoint Trajectory (HT) and Video Content information (VC). However, HT and VC are difficult to align in space due to their different modalities, and their relative importance in viewport prediction varies across prediction time steps. In this paper, we propose STAR-VP, a model that fuses HT and VC in a Space-aligned and Time-vARying manner for Viewport Prediction. Specifically, we first propose a novel saliency representation $salxyz$ and a Spatial Attention Module to solve the spatial alignment of HT and VC. Then, we propose a two-stage fusion approach based on Transformer and

gating mechanisms to capture their time-varying importance. Visualization of attention scores intuitively demonstrates STAR-VP's capability in space-aligned and time-varying fusion. Evaluation on three public datasets shows that STAR-VP achieves state-of-the-art accuracy for long-term (2-5s) viewport prediction without sacrificing short-term (<1s) prediction performance.

## CCS CONCEPTS

• **Information systems** → **Multimedia streaming**; • **Mathematics of computing** → *Time series analysis*; • **Computing methodologies** → Neural networks.

## KEYWORDS

viewport prediction, 360° video, Transformer network, multi-modal fusion, saliency map

**ACM Reference Format:**
Baoqi Gao, Daoxu Sheng, Lei Zhang, Qi Qi, Bo He, Zirui Zhuang, and Jingyu Wang. 2024. STAR-VP: Improving Long-term Viewport Prediction in 360° Videos via Space-aligned and Time-varying Fusion. In *Proceedings of the 32nd ACM International Conference on Multimedia (MM '24), October 28-November 1, 2024, Melbourne, VIC, Australia.* ACM, New York, NY, USA, 10 pages. https://doi.org/10.1145/3664647.3681268

---

[*]Corresponding author.

---

# 1 INTRODUCTION

360-degree videos capture the full surrounding environment, enabling viewers to freely change perspectives during playback for a more immersive and interactive viewing experience. The data rate of 360-degree video needs to be approximately six times higher than that of regular video to achieve a comparable angular resolution [1]. The increased data rate poses significant challenges for the streaming of 360-degree video over the internet. Therefore, tile-based adaptive streaming [36] has been proposed to stream only the tiles within the user's Field of View (FoV) at higher bitrates, thus reducing bandwidth and computing resource overhead. Nevertheless, the effectiveness of tile-based adaptive streaming relies heavily on the accuracy of viewport prediction. Accurate long-term viewport prediction helps pre-download tiles for a further future, establishing a longer buffer to cope with network fluctuations. Long-term viewport motion is mainly influenced by Historical viewpoint Trajectory (HT) and Video Content information (VC), thus requiring the fusion of these two types of information for prediction.

However, the fusion of HT and VC for viewport prediction faces two key challenges: 1) **Spatial alignment.** HT is represented as coordinate sequences, while commonly used VC information such as saliency map [26, 40, 43] and motion map [11, 12] is represented as image. This makes it difficult for the model to align the spatial positions represented by viewpoint coordinates in HT with the pixel positions in VC information. 2) **Time-varying importance.** The relative importance of HT and VC in viewpoint prediction varies across prediction time steps. Intuitively, in the short term, the viewpoint tends to continue its previous motion patterns due to inertia, thus HT is more important; whereas in the long term, the viewpoint tends to be attracted by saliency regions within the video content, thus VC is more important [31]. However, this time-varying pattern of relative importance can be very complex and cannot be described by simple rules.

The example in Fig. 1 further illustrates the necessity of considering the above two challenges. In the spatial dimension, User 1 and User 2's viewpoints are attracted by different salient regions due to their different positions, thus the viewport prediction model needs to align the spatial position information expressed in HT and VC. In the temporal dimension, User 1's viewpoint maintains its previous motion pattern in the short term due to inertia, until it is mainly attracted by salient regions in the video content in the long term. Therefore, the viewport prediction model needs to capture the time-varying importance of HT and VC.

In this paper, we propose STAR-VP, a model that fuses HT and VC in a Space-aligned and Time-vARying manner for Viewport Prediction. Specifically, to address the spatial alignment of HT and VC, we first propose a novel saliency representation $salxyz$, which represents a 2D grayscale saliency map as a collection of (3D spatial coordinates + saliency value) tuples, explicitly injecting spatial coordinate information into saliency information. Then, we design a Spatial Attention Module with a Transformer-like architecture to solve the spatial alignment problem in conjunction with $salxyz$. To address the time-varying importance of HT and VC, we propose a two-stage fusion approach. In the first fusion stage, we design a Temporal Attention Module with a Transformer-like architecture. This module enables the model to adaptively

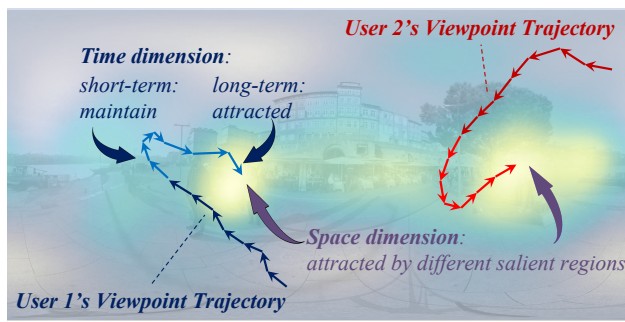

**Figure 1: An example illustrating the necessity of considering the space alignment and time-varying importance of Historical viewpoint Trajectory (HT) and Video Content information (VC) in long-term viewport prediction.**

adjust the attention to HT and VC features at each prediction step. In the second fusion stage, we design a Gating Fusion Module to further fuse the prediction results of the short-term prediction module and the long-term prediction module, enabling the model to have superior long-term prediction performance without sacrificing short-term prediction performance.

To summary, our major contributions are as follows:

- We propose a novel saliency representation $salxyz$. It explicitly injects spatial coordinate information into saliency information to facilitate the spatial alignment of HT and VC.
- We propose STAR-VP, a long-term viewport prediction model. It achieves the spatial alignment of HT and VC through a Spatial Attention Module working with $salxyz$, and captures the time-varying importance of HT and VC through a two-stage fusion approach based on Transformer and gating mechanisms.
- We evaluate STAR-VP on three public datasets and show that it achieves state-of-the-art accuracy for long-term (2-5s) viewport prediction without sacrificing short-term (<1s) prediction performance.

# 2 RELATED WORK

## 2.1 Viewport Prediction in 360° Videos

To achieve more accurate prediction, most existing viewport prediction models consider not only the user's HT, but also auxiliary information such as saliency map [26, 32, 38, 41], motion map [11, 12], object trajectory [6, 20], and other users' HT [2, 5, 25]. In this paper, we mainly explore the fusion of HT and the most commonly used VC information, saliency map.

To address the spatial alignment of HT and VC, some works [26, 34] convert HT from coordinates into VC-like heatmap representations using Gaussian filtering. While this approach unifies the modalities of HT and VC, it increases computational and storage overheads. Moreover, the spatial position information expressed in 2D ERP images cannot well reflect the motion of the viewpoint on the 3D sphere.

Regarding the time-varying importance of HT and VC, most existing viewport prediction models [11, 26, 37] ignore it and simply

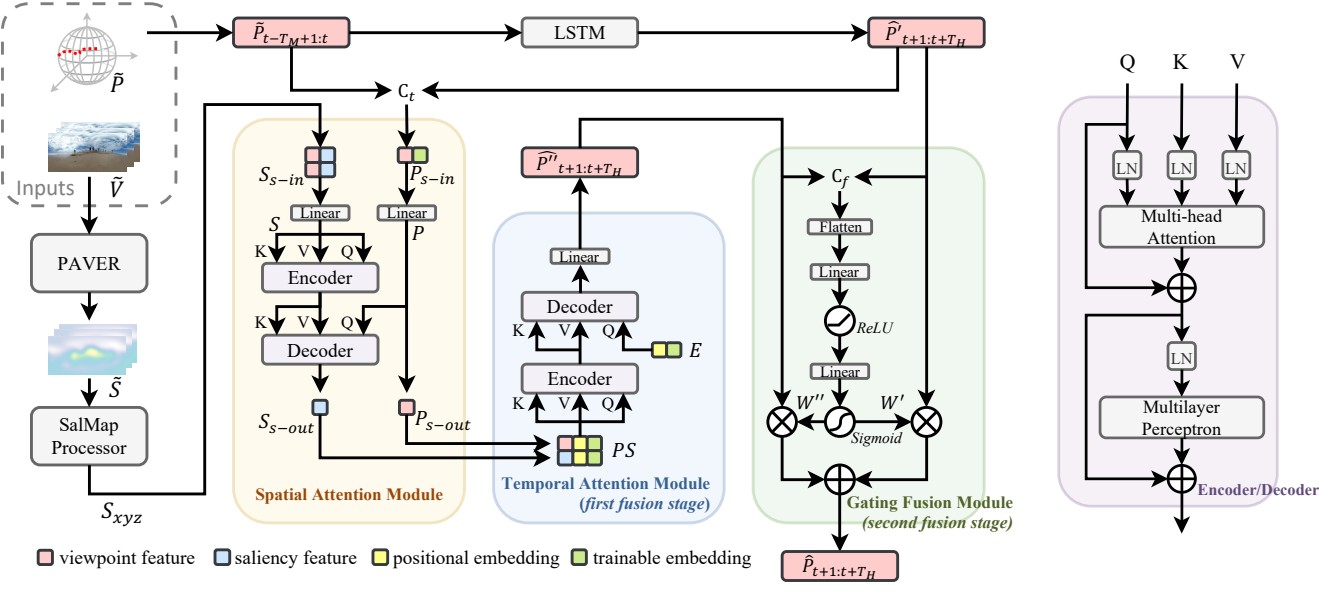

(a) STAR-VP Model Overview

(b) Encoder/Decoder Structure

**Figure 2: The architecture of STAR-VP. (a) shows the overall architecture of STAR-VP, including the SalMap Processor, LSTM
module, Spatial Attention Module, Temporal Attention Module, and Gating Fusion Module. (b) shows the details of the
Encoder/Decoder in the Spatial Attention Module and the Temporal Attention Module. The Encoder and Decoder have the same
architecture but are based on self-attention and cross-attention, respectively. $C_f(\cdot)$ and $C_t(\cdot)$ denote concatenating vectors along
the feature dimension and the temporal dimension, respectively. $\otimes$ and $\oplus$ denote element-wise multiplication and addition,
respectively. LN denotes Layer Normalization.**

concatenate HT and VC along the feature dimension. TRACK [31]
uses three LSTM modules to separately process HT, VC, and their
concatenated features, aiming to dynamically balance HT and VC
across prediction time steps. However, due to the inherent inabil-
ity of LSTMs to model complex and long-term dependencies, the
TRACK model has limitations in capturing complex time-varying
patterns of relative importance between HT and VC. HeMoG [30]
is a white-box model built on physics of rotational motion and grav-
itation, achieving higher interpretability. However, its performance
is sensitive to manually-tuned parameters. MFTR [42] uses three
Transformer encoders [33] to separately process HT, VC, and their
concatenated features. Compared to TRACK, MFTR more effec-
tively learns the time-varying importance of HT and VC. However,
MFTR still only adaptively selects HT and VC with time-varying
patterns at the feature level, making it difficult to avoid sacrific-
ing short-term prediction performance while improving long-term
prediction performance.

## 2.2 Transformer and Gating Mechanisms

Transformer [33] was initially known for its excellent performance
in natural language processing tasks such as machine translation.
It aggregates information across the entire sequence with adaptive
weights through attention mechanisms, capturing long-term depen-
dencies in a parallel manner. In recent years, models based on the
Transformer architecture have shown excellent performance not

only in natural language processing tasks [9, 16, 28] but also in var-
ious visual tasks [3, 10, 17, 21, 22] and multimodal tasks [18, 19, 23].
Since the fusion of HT and VC involves time series and multimodal
data, it is suitable to use Transformer-based models for processing.

The application of gating mechanisms in neural networks can
be traced back to the LSTM model [14]. LSTM uses three gating
units to control the flow of information to better capture long-term
dependencies. Since then, gating mechanisms have been widely
used in various deep learning models [7, 27, 45] to better focus on
important information and suppress irrelevant information.

## 3 METHOD

### 3.1 Overview

The overall architecture of STAR-VP is depicted in Fig. 2. STAR-VP
predicts the viewport positions for the future $T_H$ time steps($\widehat{P}$),
given the viewport positions from the past $T_M$ time steps ($\widetilde{P}$) and
the video frames spanning both the past $T_M$ and the future $T_H$ time
steps ($\widetilde{V}$). Each viewport position is represented as a 3D spatial
coordinate on the unit sphere, and each video frame is represented
as an image in Equi-Rectangular Projection (ERP) format.

STAR-VP mainly consists of five modules: SalMap Processor,
LSTM module, Spatial Attention Module, Temporal Attention Mod-
ule, and Gating Fusion Module. First, the original video frames $\widetilde{V}$
are converted into saliency maps $\widetilde{S}$ by PAVER [39], a model specifi-
cally designed for saliency detection in 360-degree videos. Then, the
SalMap Processor converts $\widetilde{S}$ into the *salxyz* representation $S_{xyz}$.

The LSTM module only uses the viewpoint positions from the past $T_M$ time steps to obtain predictions $\widehat{P'}$ with excellent short-term performance for the future $T_H$ time steps. The Spatial Attention Module takes $S_{xyz}$ and the concatenation of the historical viewpoint trajectory and the LSTM prediction results as input, and outputs the spatially aligned viewpoint position features $P_{s-out}$ and saliency features $S_{s-out}$. The Temporal Attention Module is responsible for the first-stage fusion. It takes $P_{s-out}$ and $S_{s-out}$ as input, adaptively adjusts the focus on HT and VC features at each prediction time step through well-designed attention modules, and outputs predictions $\widehat{P''}$ with excellent long-term performance. The Gating Fusion Module is responsible for the second-stage fusion. It uses a gating mechanism to obtain the weights of $\widehat{P'}$ and $\widehat{P''}$ at each prediction time step, and outputs the final viewpoint prediction results $\widehat{P}$.

The following subsections will introduce the specific details of the five modules mentioned above and the internal Encoder/Decoder.

## 3.2 SalMap Processor

The SalMap Processor converts the original saliency map into a compact saliency representation with pixel coordinate information. Given the saliency map $\widetilde{S} \in \mathbb{R}^{(T_M+T_H) \times H \times W \times 1}$, the SalMap Processor represents each pixel's saliency information as $(x, y, z, s)$, where $(x, y, z)$ are the 3D coordinates of the pixel block center and $s$ is the corresponding saliency value. The 3D coordinates $(x, y, z)$ of the pixel block with index $(i, j)$ are calculated as follows:

$$(x, y, z) = (\cos\theta \cdot \sin\phi, \ \sin\theta \cdot \sin\phi, \ \cos\phi),$$
$$\theta = \frac{2\pi}{W} \cdot (j + 0.5), \ \phi = \frac{\pi}{H} \cdot (i + 0.5), \quad (1)$$

where $H$ and $W$ are the height and width of the saliency map respectively.

To reduce data redundancy, the SalMap Processor retains only the top $tt$ points with maximum saliency, which are then sampled at rate $sr$. Finally, the SalMap Processor outputs $S_{xyz} \in \mathbb{R}^{(T_M+T_H) \times D_P \times 4}$, where $D_P = \lfloor tt \cdot sr \rfloor$.

## 3.3 LSTM Module

When watching 360-degree videos, users' viewpoint motion is relatively stable most of the time. Due to inertia, the future motion of the viewpoint, especially in the short term, is highly correlated with its historical trajectory. Therefore, we use LSTM, a lightweight and efficient temporal prediction model, to take only the viewpoint positions from the past $T_M$ time steps as input and output predictions for the future $T_H$ time steps in an autoregressive manner:

$$\widehat{P'}_{t+1:t+T_H} = \text{LSTM}(\widetilde{P}_{t-T_M+1:t}), \quad (2)$$

where $\widetilde{P}_{t-T_M+1:t} \in \mathbb{R}^{T_M \times 3}$ is the viewpoint positions from the past $T_M$ time steps, and $\widehat{P'}_{t+1:t+T_H} \in \mathbb{R}^{T_H \times 3}$ is the LSTM prediction results for the future $T_H$ time steps. This prediction has excellent short-term performance. The specific details of the LSTM module are omitted here.

## 3.4 Spatial Attention Module

The Spatial Attention Module is based on a Transformer-like architecture. Compared to the original Transformer model, we have

modified its positional encoding part and the structure and query input of its Decoder.

First, since both $S_{xyz}$ and the viewpoint data already contain 3D spatial coordinate information, we modify the positional encoding in the original Transformer to trainable parameter embedding, aiming to unify the representations of the two modalities. The Spatial Attention Module takes $S_{s-in}$ and $P_{s-in}$ as input, where $S_{s-in} = S_{xyz} \in \mathbb{R}^{(T_M+T_H) \times D_P \times 4}$. The calculation formula of $P_{s-in}$ is as follows:

$$P_{s-in} = \mathbf{C}_f(\mathbf{C}_t(\widetilde{P}_{t-T_M+1:t}, \widehat{P'}_{t+1:t+T_H}), TE) \in \mathbb{R}^{(T_M+T_H) \times 4}, \quad (3)$$

where $\mathbf{C}_f(\cdot)$ and $\mathbf{C}_t(\cdot)$ denote concatenating vectors along the feature dimension and the temporal dimension respectively, and $TE \in \mathbb{R}^{(T_M+T_H) \times 1}$ is a vector expanded from a $1 \times 1$ trainable parameter along the temporal dimension, used to transform $P_{s-in}$ into a form similar to $(x, y, z, s)$ of $S_{s-in}$, facilitating modality fusion and spatial alignment. Subsequently, two linear layers convert $S_{s-in}$ and $P_{s-in}$ into $S \in \mathbb{R}^{(T_M+T_H) \times D_P \times D_C}$ and $P \in \mathbb{R}^{(T_M+T_H) \times D_C}$, respectively.

Next, we unify the Decoder to have the same architecture as the Encoder, and use viewpoint features as the query input to the Decoder, fully utilizing the cross-attention mechanism to decode viewpoint-specific saliency features. The Encoder encodes $S$ into $S'$ using self-attention, with $S$ serving as query, key, and value. The Decoder uses viewpoint position features $P$ as query and $S'$ as key and value, employing cross-attention to perceive the position information of viewpoint, giving higher attention to the saliency information near the viewpoint. Further details of the Encoder/Decoder are provided in Section 3.7.

## 3.5 Temporal Attention Module

Similar to the Spatial Attention Module, the Temporal Attention Module is also based on a Transformer-like architecture. Compared to the original Transformer model, we have modified its positional encoding part and the structure and query input of its Decoder.

First, we modify the positional encoding in the original Transformer to positional embedding and trainable modality-specific embedding. The Temporal Attention Module takes $P_{s-out} = P \in \mathbb{R}^{(T_M+T_H) \times D_C}$ and $S_{s-out} \in \mathbb{R}^{(T_M+T_H) \times D_C}$ as input, and outputs the composite vector $PS \in \mathbb{R}^{2(T_M+T_H) \times D_{PS}}$ as follows:

$$PS = \mathbf{C}_t(\mathbf{C}_f(P_{s-out}, PE, TE_P), \mathbf{C}_f(S_{s-out}, PE, TE_S)), \quad (4)$$

where $\mathbf{C}_f(\cdot)$ and $\mathbf{C}_t(\cdot)$ denote concatenating vectors along the feature dimension and the temporal dimension respectively, $PE \in \mathbb{R}^{(T_M+T_H) \times D_{PE}}$ is fixed 1D Fourier positional embeddings for a total of $T_M + T_H$ time steps, $TE_P \in \mathbb{R}^{(T_M+T_H) \times D_{TE}}$ and $TE_S \in \mathbb{R}^{(T_M+T_H) \times D_{TE}}$ are trainable modality-specific embeddings, and $D_{PS} = D_C + D_{PE} + D_{TE}$.

Next, we use the vector concatenated by positional embeddings and modality-specific embeddings as the query input to the Decoder, fully utilizing the cross-attention mechanism to decode the viewpoint information at each prediction step. The query embeddings $E \in \mathbb{R}^{T_H \times (D_{PE}+D_{TE})}$ are calculated as follows:

$$E = \mathbf{C}_f(PE_{t+1:t+T_H}, TE_{P \ t+1:t+T_H}), \quad (5)$$

The Decoder in the Temporal Attention Module is also unified to have the same architecture as the Encoder. The Encoder encodes

$PS$ into $PS'$ using self-attention, with $PS$ serving as query, key, and value. The Decoder uses $E$ as query and $PS'$ as key and value, employing cross-attention to perceive the specific temporal and modality information. This captures the time-varying importance of the viewpoint position features and saliency features. The Encoder/Decoder is introduced in detail in Section 3.7.

As the first fusion stage, the Temporal Attention Module adaptively adjusts the focus on HT and VC features at each prediction time step, outputting predictions $\widehat{P''} \in \mathbb{R}^{T_H \times 3}$ with excellent long-term performance.

## 3.6 Gating Fusion Module

As the second fusion stage, the Gating Fusion Module generates vectors $W' \in \mathbb{R}^{T_H \times 1}$ and $W'' \in \mathbb{R}^{T_H \times 1}$ to control the weights of $\widehat{P'}$ and $\widehat{P''}$ at each prediction time step, and outputs the final viewpoint prediction results $\widehat{P} \in \mathbb{R}^{T_H \times 3}$:

$$
\begin{aligned}
\widehat{P} &= W' \otimes \widehat{P'} \oplus W'' \otimes \widehat{P''}, \quad x = \text{Flatten}(\mathbf{C}_f(\widehat{P'}, \widehat{P''})), \\
W' &= \sigma_s(W_s(\sigma_r(W_r x + b_r)) + b_s), \quad W'' = 1 - W'
\end{aligned}
\tag{6}
$$

where $\otimes$ and $\oplus$ denote element-wise multiplication and addition respectively, $\mathbf{C}_f$ denotes concatenating vectors along the feature dimension, $\sigma_r$ is the ReLU activation function, $\sigma_s$ is the sigmoid activation function, $x \in \mathbb{R}^{6T_H \times 1}$, $W_r \in \mathbb{R}^{D_G \times 6T_H}$, $b_r \in \mathbb{R}^{D_G \times 1}$, $W_s \in \mathbb{R}^{T_H \times D_G}$, $b_s \in \mathbb{R}^{T_H \times 1}$.

## 3.7 Encoder and Decoder

Inspired by the universal multimodal fusion architecture Perceiver IO [15], we unified the architecture of the Encoder and Decoder in the Spatial Attention Module and the Temporal Attention Module, and added some additional linear layers to adjust the feature dimensions of the vectors. The Encoder uses self-attention to extract deep features, while Decoder uses cross-attention to perceive current specific spatial/temporal information, achieving adaptive feature selection. The Encoder/Decoder consists of two components: the Multi-head Attention and the Multilayer Perceptron (MLP).

The Multi-head Attention component performs the following calculation given $Q_{in} \in \mathbb{R}^{n_q \times d_q}$, $K_{in} \in \mathbb{R}^{n_k \times d_k}$, and $V_{in} \in \mathbb{R}^{n_v \times d_v}$:

$$
\begin{aligned}
\text{Attn}(Q_{in}, K_{in}, V_{in}) &= \mathcal{L}_o(\text{softmax}(\frac{Q'K'^T}{\sqrt{d_l}})V'), \\
Q' &= \mathcal{L}_q(Q_{in}), \quad K' = \mathcal{L}_k(K_{in}), \quad V' = \mathcal{L}_v(V_{in}),
\end{aligned}
\tag{7}
$$

where $\mathcal{L}_{q,k,v}$ are linear layers mapping each input to a shared feature dimension $d_l$, and $\mathcal{L}_o$ is a linear layer projecting the output to a desired feature dimension. It is worth noting that all of these linear layers are applied convolutionally along the index dimension, and the batch and head dimensions have been excluded for clarity.

The MLP component consists of two linear layers with a GELU [13] nonlinearity applied after the first layer:

$$
\text{MLP}(X_{in}) = \mathcal{L}_2(\text{GELU}(\mathcal{L}_1(X_{in})))
\tag{8}
$$

The final output of the Encoder/Decoder is calculated as follows:

$$
\begin{aligned}
\text{Attention}(Q, K, V) &= \mathcal{A}(O_{attn}, \text{MLP}(\mathcal{N}(O_{attn}))), \\
O_{attn} &= \mathcal{A}(Q, \text{Attn}(\mathcal{N}(Q), \mathcal{N}(K), \mathcal{N}(V))),
\end{aligned}
\tag{9}
$$

where $\mathcal{N}$ denotes layer normalization, and $\mathcal{A}$ denotes addition in residual connection.

## 4 EXPERIMENTS

STAR-VP is evaluated on a unified evaluation framework [29] for viewport prediction methods in 360-degree videos. The framework unifies five datasets into a common format (sampling period: 0.2s; viewpoint representation: 3D coordinates on the unit sphere) and summarizes current combination approaches of HT and VC, evaluating their performance.

### 4.1 Setup

*4.1.1 Datasets.* We evaluate STAR-VP on three datasets:
(1) *David_MMSys_18* [8] contains 19 20-second 360-degree videos in ERP format and head and eye tracking data from 57 participants watching the videos freely.
(2) *Nasrabadi_MMSys_19* [24] consists of 28 videos (60s) and viewport traces from 60 participants watching the videos. Each video is viewed by 30 participants.
(3) *Wu_MMSys_17* [35] includes head tracking data from 48 users watching 18 360-degree videos of 5 categories.

*4.1.2 Competitors.* Five models are implemented as competitors. Three of them only use HT:
(1) *linear-regression* predicts future viewpoint positions using a linear regression model.
(2) *deep-pos-only* is an LSTM-based encoder-decoder model. The encoder receives the historical viewpoint trajectory and generates an internal representation. The decoder receives the output of the encoder and progressively produces predictions over the target horizon, by re-injecting the previous prediction as input for the new prediction time-steps.
(3) *VPT360* [4] is a Transformer-based model. It only uses Transformer Encoder to process HT information for time series prediction.
The other two models use both HT and VC:
(4) *TRACK* [31] uses three LSTM modules to separately process HT, VC, and their concatenated features, aiming to dynamically balance HT and VC across prediction time steps.
(5) *MFTR* [42] is a complex multi-modal fusion Transformer-based model. However, its basic fusion framework for HT and VC is to use three Transformer-encoder-based modules to separately process HT, VC, and their concatenated features.

It is worth noting that the above comparison models basically cover all the fusion methods of HT and VC in the current viewport prediction models we know.

*4.1.3 Performance Metrics.* We use Orthodromic Distance (OD) [31] and Intersection over Union (IoU) [44] as the performance metrics.

The Orthodromic Distance between points $P1$ and $P2$ is calculated as follows:

$$
OD(P_1, P_2) = \arccos(\vec{P_1} \cdot \vec{P_2})
\tag{10}
$$

where $P_1$ and $P_2$ are two 3D coordinates on the unit sphere, and $\cdot$ is the dot product operation. When we use "3D spatial coordinates on the unit sphere" to represent the viewpoint position, the orthodromic distance actually represents the shortest spherical distance between the two viewpoints.

The Intersection over Union is an evaluation metric related to the specific tile partition method. We divide the ERP video frame into $9 \times 16$ tiles and set the user's field of view to $100°$. If the angle between the center of the tile and the viewpoint is less than $50°$, the tile is considered to be in the viewport, and its label is set to 1; otherwise, it is set to 0. Then, the Intersection over Union is computed on these labels as follows:

$$IoU = \frac{TP}{TT} \tag{11}$$

where $TP$ represents *True Positive*, i.e., the intersection between prediction and ground-truth of tiles with label 1; $TT$ represents *True Total*, i.e., the union of all tiles with label 1 in the prediction and in the ground-truth.

*4.1.4 Saliency Map Generation and Processing.* We use PAVER [39], a Vision Transformer-based model trained specifically for 360° video saliency detection, to compute the saliency map at each time step. The original $224 \times 448$ saliency maps are processed by the SalMap Processor module of STAR-VP to generate $128 \times 4$ compact saliency representation $S_{xyz}$ with a top threshold ($tt$) of 20480 and sample rate ($sr$) of 1/160. In addition, for model size and memory considerations, the competitors use downsampled $64 \times 128$ saliency maps as the actual VC input. Even so, the data volume of saliency information actually used by STAR-VP is only 1/16 of that of other competitors.

*4.1.5 Model Hyperparameters.* The hyperparameters of all models are shown in Table 1. All models predict the next 5s (25 steps) of viewpoint positions based on the past 3s (15 steps).

**Table 1: Hyperparameters of all models.**

| Symbol | Value | Description |
|---|---|---|
| $D_{cL}$ | 256 | Channel dimension of the hidden state in LSTM module. |
| $D_{cT}$ | 256 | Channel dimension of input of the Transformer encoder. |
| $D_{PE}$ | 129 | Dimension of the positional embedding. |
| $D_{TE}$ | 127 | Dimension of the trainable embedding. |
| $D_G$ | 128 | Dimension of the hidden state in the Gating Fusion Module. |
| $N_{layers^L}$ | 2 | Number of layers in LSTM module. |
| $N_{layers^T}$ | 2 | Number of self-attention layers per block. |
| $N_{blocks}$ | 2 | Number of blocks in the Transformer encoder. |
| $N_{heads}$ | 8 | Number of heads in the multi-head attention layer. |

**Table 2: The average long-term prediction (2-5s) performance of viewport prediction models on three datasets. OD is Orthodromic Distance, and IoU is Intersection over Union.**

| | David_MMSys_18 | | Nasrabadi_MMSys_19 | | Wu_MMSys_17 | |
| | OD ↓ | IoU ↑ | OD ↓ | IoU ↑ | OD ↓ | IoU ↑ |
|---|---|---|---|---|---|---|
| linear-regression | 1.174 | 23.92% | 0.962 | 35.41% | 0.643 | 51.17% |
| deep-pos-only | 1.145 | 26.35% | 0.950 | 35.93% | 0.623 | 52.37% |
| TRACK | 1.123 | 25.05% | 0.939 | 35.05% | 0.613 | 51.12% |
| VPT360 | 1.127 | 26.00% | 0.941 | 36.27% | 0.624 | 52.04% |
| MFTR | 1.064 | 27.98% | 0.954 | 34.27% | 0.599 | 52.02% |
| **STAR-VP (ours)** | **0.967** | **33.26%** | **0.862** | **39.84%** | **0.531** | **56.82%** |

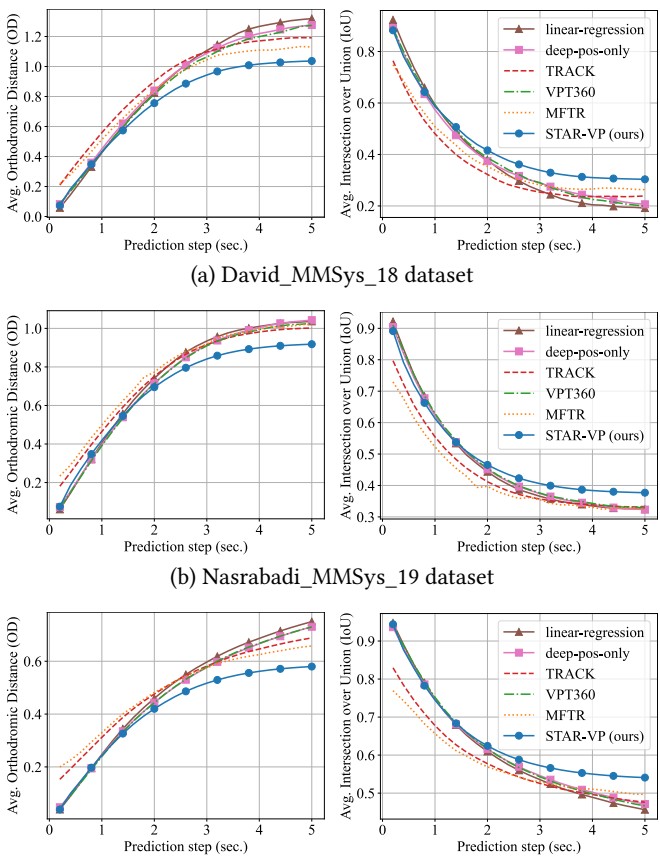

(a) David_MMSys_18 dataset

(b) Nasrabadi_MMSys_19 dataset

(c) Wu_MMSys_17 dataset

**Figure 3: The Orthodromic Distance (OD) and Intersection over Union (IoU) against prediction steps of viewport prediction models on each dataset.**

## 4.2 Results and Analysis

Fig. 3 shows the prediction performance of STAR-VP and five competitors on three datasets. From the figure, we can see that HT-only models (linear-regression, deep-pos-only, VPT360) have better performance in short-term prediction (<1s) as they are not influenced by VC information. The two competitors (TRACK, MFTR) that consider both HT and VC for prediction have better long-term prediction (2-5s) performance, but worse short-term prediction performance than HT-only models. This indicates that their improvement in long-term prediction performance comes at the expense of short-term prediction performance. In contrast, STAR-VP achieves the best long-term prediction performance without sacrificing short-term prediction performance. This is mainly due to the space-aligned and time-varying fusion of HT and VC by using *salxyz* representation and the two-stage fusion method in STAR-VP. Table 2 further shows the average long-term prediction performance of each model on three datasets. We note that because TRACK and MFTR sacrifice short-term prediction performance to improve long-term prediction, their average performance over 2-5s is often worse than HT-only models such as deep-pos-only and

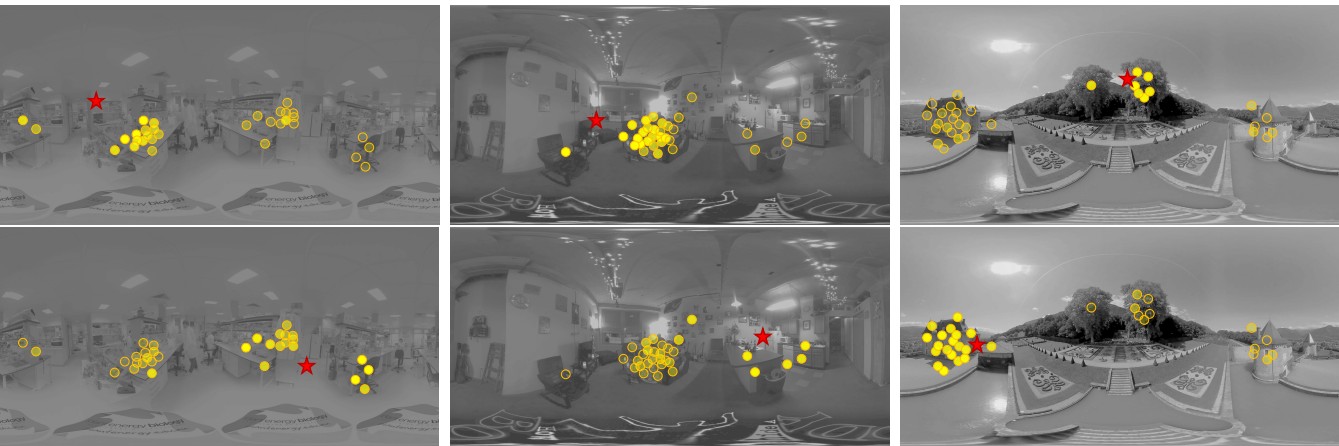

**Figure 4: The visualization of the spatial attention scores assigned by STAR-VP to the saliency sampling points (yellow points) on three video frames, given two different user viewpoint positions (red star). The brighter the yellow point, the larger the spatial attention score.**

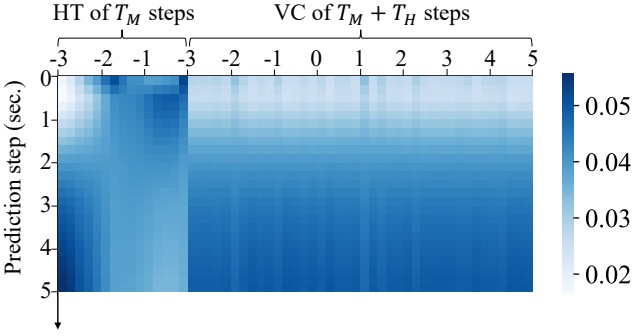

**Figure 5: The heatmap of the column-normalized temporal attention scores between each prediction step and the HT/VC inputs from STAR-VP.**

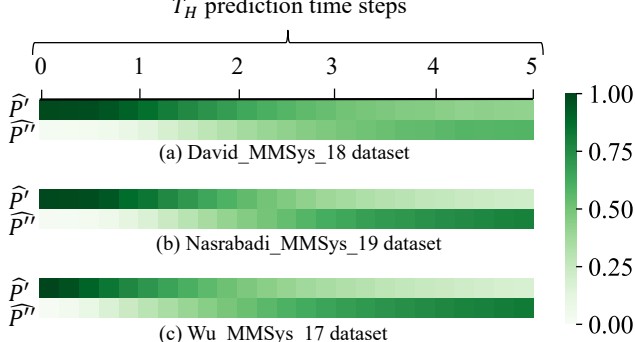

**Figure 6: The heatmap of the weight allocation of the Gating Fusion Module in STAR-VP for $\widehat{P'}$ and $\widehat{P''}$ at each prediction time step on three datasets.**

VPT360. Across all datasets and metrics, STAR-VP achieves state-of-the-art performance in long-term prediction and comparable performance in short-term prediction.

## 4.3 Visualization

*4.3.1 Attention Scores of Spatial Attention Module.* To analyze STAR-VP's ability to align HT and VC spatially, we multiply the attention matrices of its Spatial Attention Module to obtain the attention scores between the viewpoint and each saliency sampling point on video frames and visualize them as shown in Fig. 4. The figure shows how the attention scores of the saliency sampling points on three different video frames change with the spatial position of the current user viewpoint. The red star represents the current user's viewpoint on the video frame, and the yellow points are 32 points uniformly sampled from the *salxyz* representation of the current frame according to saliency value. The brighter the yellow point, the higher the attention score. It is evident from the figure that STAR-VP assigns higher attention scores to the saliency

sampling points closer to the viewpoint. This indicates that it aligns the spatial information expressed by HT and VC well.

*4.3.2 Attention Scores of Temporal Attention Module.* To analyze the time-varying attention pattern of STAR-VP, we multiply the attention matrices of its Temporal Attention Module to obtain the attention scores between each prediction step and the HT/VC inputs, and visualize them as the heatmap shown in Fig. 5. It can be seen from the figure that as the prediction step increases, the attention scores of recent 2s HT gradually decrease, while the attention scores of VC and earlier HT gradually increase. This is consistent with the idea that HT reflects inertia for short-term viewpoint motion, while VC influences long-term motion through salient regions. We also observe that as the prediction step increases, STAR-VP gradually increases its attention to HT over the past 2s. This is because HT over the past 2s reflects certain user viewpoint motion preferences that are beneficial for long-term viewpoint prediction. In summary, the visualization results of the attention scores of

the Temporal Attention Module indicate that STAR-VP performs feature selection on HT and VC with a time-varying pattern in the first fusion stage. It is worth noting that to facilitate a clearer observation of the time-varying attention pattern, we apply min-max normalization to the original attention scores matrix along the columns.

*4.3.3  Weight Allocation of Gating Fusion Module.* We visualize the weight allocation of the Gating Fusion Module in STAR-VP for $\widehat{P'}$ and $\widehat{P''}$ at each prediction time step as a heatmap, as shown in Fig. 6. Here, $\widehat{P'}$ is the prediction result with better short-term performance output by the LSTM module that only considers HT, while $\widehat{P''}$ is the prediction result with better long-term performance output by the Temporal Attention Module after the first-stage fusion of HT and VC. It can be seen from the figure that for the initial prediction steps, the Gating Fusion Module almost assigns all the weight to $\widehat{P'}$. As the prediction step increases, the weight of $\widehat{P'}$ is gradually reduced, while that of $\widehat{P''}$ is increased. This allows STAR-VP to achieve excellent long-term prediction performance without sacrificing short-term performance.

## 4.4  Ablation Study

*4.4.1  Effects of Each Module in STAR-VP.* To analyze the contribution of each module in STAR-VP to the prediction performance, we conducted an ablation study on the Spatial Attention Module (SAM), Temporal Attention Module (TAM), and Gating Fusion Module (GFM). For SAM, we replaced it with a linear layer. For TAM, we replaced it with an LSTM. For GFM, we removed it directly and used the output of TAM as the final prediction result. The results are depicted in Fig. 7. It can be seen from the figure that removing any module will affect the prediction performance. Removing SAM and TAM mainly affects long-term prediction performance, while removing GFM mainly affects short-term performance. This result not only demonstrates the importance of each module but also indicates that the main role of SAM and TAM is to improve the long-term viewpoint prediction performance of STAR-VP, while the main role of GFM is to block the output of the long-term prediction module in short-term prediction to ensure the short-term viewpoint prediction performance of STAR-VP.

*4.4.2  Effects of salxyz Representation.* We tested the long-term prediction performance of three models that consider both HT and VC with the original saliency map and the *salxyz* representation,

**Table 3: Ablation study results of *salxyz*: the long-term prediction (2-5s) performance of three models considering HT and VC with or without the *salxyz* representation. OD is Orthodromic Distance, and IoU is Intersection over Union.**

|  | David_MMSys_18 | | Nasrabadi_MMSys_19 | | Wu_MMSys_17 | |
|---|---|---|---|---|---|---|
|  | OD ↓ | IoU ↑ | OD ↓ | IoU ↑ | OD ↓ | IoU ↑ |
| TRACK (w/o *salxyz*) | 1.123 | 25.05% | 0.939 | 35.05% | 0.613 | 51.12% |
| TRACK (w/ *salxyz*) | 1.144 | 24.62% | 0.952 | 34.35% | 0.626 | 50.90% |
| MFTR (w/o *salxyz*) | 1.064 | 27.98% | 0.954 | 34.27% | 0.599 | 52.02% |
| MFTR (w/ *salxyz*) | 1.123 | 27.34% | 0.958 | 34.02% | 0.613 | 52.03% |
| STAR-VP (w/o *salxyz*) | 1.057 | 28.31% | 0.928 | 36.91% | 0.584 | 52.56% |
| **STAR-VP** (w/ *salxyz*) | **0.967** | **33.26%** | **0.862** | **39.84%** | **0.531** | **56.82%** |

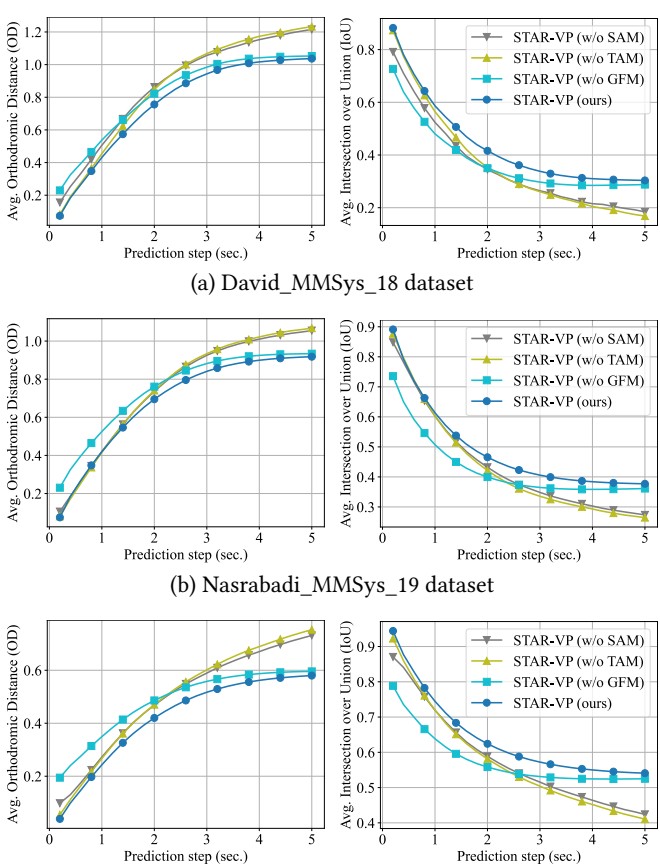

(a) David_MMSys_18 dataset

(b) Nasrabadi_MMSys_19 dataset

(c) Wu_MMSys_17 dataset

**Figure 7: Ablation study results of Spatial Attention Module (SAM), Temporal Attention Module (TAM), and Gating Fusion Module (GFM) in STAR-VP on each dataset.**

respectively. The results are shown in Table 3. From the test results, it can be seen that the performance of the TRACK and MFTR models decreases after using the *salxyz* representation, while only STAR-VP improves its performance after using the *salxyz* representation. This result demonstrates the improvement of the *salxyz* representation on long-term prediction performance and highlights the importance of the *salxyz* representation in conjunction with the Spatial Attention Module.

## 5  CONCLUSION

In this paper, we propose a novel long-term viewpoint prediction model STAR-VP. It fuses viewpoint and saliency information in a space-aligned and time-varying manner, achieving the best long-term prediction performance without sacrificing short-term prediction performance. A viewpoint prediction model with excellent long-term performance helps 360° video streaming systems pre-download tiles for a longer time ahead, thereby establishing a longer buffer to cope with network fluctuations.

## ACKNOWLEDGMENTS

This work was supported by the National Natural Science Foundation of China under Grants (U23B2001, 62101064, 62171057, 62201072, 62001054, 62071067), the Ministry of Education and China Mobile Joint Fund (MCM20200202, MCM20180101), Beijing University of Posts and Telecommunications-China Mobile Research Institute Joint Innovation Center, China Postdoctoral Science Foundation (2023TQ0039). National Postdoctoral Program for Innovative Talents under Grant BX20230052.

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
