# OpenReview forum: "STAR-VP: Improving Long-term Viewport Prediction in 360° Videos via Space-aligned and Time-varying Fusion"
_acmmm.org/ACMMM/2024/Conference — MM2024 Poster_

### Official Review · Reviewer_7nmo · 2024-05-21

**Rating:** 5
**Confidence:** 4

**Summary:**

This paper proposes an approach to improve the accuracy of viewport prediction for 360-degree video streaming. The approach takes both historical viewpoint trajectories and video content as input, and predicts the viewpoint in the future 1-5 seconds. Existing works either perform well in the short term (< 1 s) or long term (2 - 5 s). The proposed method uses a transformer-like mechanism to conduct spatial and temporal fusions, achieving higher prediction accuracy in both the short and long term, compared to existing techniques.

**Strengths:**

The paper addresses an important problem in 360-degree video streaming, and provides a technically sound solution. It is well-known that both historical viewpoint trajectories and video content are useful for viewport prediction, but how to adaptively balance their contributions over time is a challenging issue. This paper presents a promising solution to this problem. The evaluation is comprehensive, conducted on three datasets and using two evaluation metrics. The visualizations and ablation study are clear and helpful.

**Limitations:**

While the proposed method outperforms existing techniques, especially in the longer term, the overall performance may not be significant enough to make it impactful in practice. For example, in the short term, the method's performance is similar to simpler approaches like linear regression. In the longer term (2-5 seconds), the IoU metric is around 0.3, which may still be insufficient for practical deployment.

The writing could be improved in a few places:
1) In Section 1, the second paragraph states that spatial alignment is hard, but it would be more logical to first explain why spatial alignment is necessary for the problem before discussing the challenges. The authors explain why in the third paragraph, which does not look right.
2) The dimensions of the Ss-out and Ps-out tensors are not explicitly mentioned in Section 3.5. Adding this information would help readers better understand the model architecture.

**Suitability:**

2

---

### Official Review · Reviewer_Ts5H · 2024-05-24

**Rating:** 2
**Confidence:** 3

**Summary:**

This paper proposed a framework that combines saliency features with head movement history to predict long-term head movement.

**Strengths:**

The paper employs an attention structure to match two modalities: head movement history and saliency content. Attention is a well-studied building block with proven performance.

**Limitations:**

The paper used five different components to perform long-term viewport prediction. The structure is complex but lacks clarification. Specifically:
+Why did the authors consecutively stack encoders and decoders? Is there any empirical or mathematical evidence to prove the superiority of this approach?
+It is true that LSTM cannot handle long-term prediction. Then why incorporate LSTM into the structure as it would introduce biases and increase prediction errors? Isn't using the attention structure alone sufficient, as it can handle both long-term and short-term information?


The author states that the goal of the paper is to support 'real-time streaming of 360-degree video,' but this goal is difficult to achieve as the model is computationally expensive. It first uses SalMap Processor to extract saliency maps for every video frame, then uses multiple attention building blocks.

**Suitability:**

2

---

### Official Review · Reviewer_4Wsg · 2024-05-24

**Rating:** 5
**Confidence:** 3

**Summary:**

The paper proposes a novel viewport prediction algorithm for 360 degree video that historical trajectory and video content information through a Transformer architecture. In addition, a novel saliency representation salxyz is proposed. Its performance and generalizability are shown on three different, public datasets indicating its improvement for long-term prediction horizons.

**Strengths:**

- The paper is well-written, has a good structure and is fluent to follow. Its contributions are novel, relevant and timely.
- The authors propose an interesting line of thought by linking the applicability of content-agnostic vs content-aware prediction to the envisioned prediction horizon.
- The performance of the model is proven on multiple, publicly accessible and well-known viewport prediction datasets.

**Limitations:**

- The stated problem regarding spatial alignment of movement trajectories with content information is not entirely clear to me. Additional explanation is required to this extent.
- I don't fully understand the need for a novel saliency representation. How are other saliency methods represented? How does salxyz differs from these?
- It might be interesting to include static prediction (i.e. always take the last known orientation as the prediction) as a baseline, as it incorporates the most straightforward prediction possible.
- The authors state that data is randomly divided in training, validation and test set at user level. Does this mean that trajectories from different users but a same, given video might appear in both training and validation/test set? How does this affect the results? How would the model perform on a previously unseen video?

**Suitability:**

3

---

### Official Review · Reviewer_1a8L · 2024-06-04

**Rating:** 5
**Confidence:** 3

**Summary:**

The paper proposes a model called STAR-VP that improves long-term viewport prediction in 360° videos through space-aligned and time-varying fusion of historical viewpoint trajectory and video content information leveraging a transformer architecture.

**Strengths:**

-	The idea of aligning the spatial information and the saliency information for further processing is interesting and seems to work well in practice
-	The approach is well thought and improves compared to sota
-	The evaluation is performed using a common evaluation framework, meaning that the results should be easier to validate and use in future works as well

**Limitations:**

-	Even though interesting, not sure if the spatial representation is big enough to be claimed as a contribution to the paper
-	All of the proposed modules seem to contribute towards better performance for long term prediction compared to sota. What is unclear though is exactly how the relationship between short term prediction (dominated by the user trajectory) and long term one (dominated by the saliency map) plays out. A qualitative analysis could have been helpful to shed some light on this aspect

**Suitability:**

3

---

### Meta-Review · Area_Chair_bAb9 · 2024-06-30

**Recommendation:** Accept (Poster)
**Confidence:** 4

**Metareview:**

This paper proposes an approach for viewport prediction for 360-degree video streaming which takes viewport trajectories and video content into approach. The paper is well written, relevant and interesting. The reviewers in general like the paper and are inclined to its acceptance. However, after the rebuttal, more questions regarding the novelty contribution were risen, leading to some of reviewers to low their score to borderline accept. Thus, while I see the value of the paper, I would recommend it as a poster.